# A Multi-load balancing control strategy for a novel low carbon integrated energy system for buildings

Wenjie Wang[1], Yefan Shu[1], Zhengyu Wang[2], Chenliang Ji[1], Hui Huang[1], Rundong Ji[3], Mengxiong Zhou[1], Yaodong Wang[4], Jie Ji[1]*

1 Huaiyin Institute of Technology, Huaiyin Jiangsu, China, 2 Peking University, Beijing, China, 3 Jiangsu Huashui Engineering Detection & Consulting Co. Ltd, Huai'an, Jiangsu, China, 4 Durham Energy Institute Durham University, Durham, England

* jijie@hyit.edu.cn

## Abstracts

With the escalating global energy demand and the pressing need for clean energy, innovations in building energy systems are crucial. This paper proposes a Multi - load Balancing Control Strategy (MLBS) based on the Improved Tuna Swarm Optimization (ITSO) algorithm. By integrating battery and compressed air energy storage, the strategy enhances the building energy system's flexible dispatch capability. A building complex in Huai'an, Jiangsu is used as a case study. Two cases are compared: one without MLBS and the other with MLBS using a low - carbon economic dispatch model. Results show that MLBS effectively adjusts electric and thermal loads. After applying MLBS and the low - carbon economic dispatch strategy, the quarterly planning cost drops to 110.37 million, the operating cost to 204.28 tons, the carbon trading cost to 8.15 million, and the total carbon emission to 137.27 million, significantly lower than the values without the strategy. Moreover, the TSO - MLBS energy scheduling strategy adopted in this paper has the shortest computation time and lower energy scheduling cost, offering a double improvement in economic efficiency and carbon emission reduction.

## 1 Introduction

Buildings account for significant global energy consumption in total carbon emissions [1]. As urbanization continues, building energy demand will increase by 50% over 30 years [2]. At the same time, an unprecedented energy revolution is taking place globally, and how to ensure a sustainable supply of energy while effectively controlling energy prices and reducing greenhouse gas (GHG) emissions from energy use is a common concern for countries around the world today. In September 2020, China put forward the "double carbon" target of "peak carbon" by 2030 and "carbon neutrality" by 2060; more and more distributed renewable energy sources will be connected to

**Data availability statement:** All relevant data are within the paper.

**Funding:** The author(s) received no specific funding for this work.

**Competing interests:** The authors have declared that no competing interests exist.

buildings and distribution systems, which will bring more severe challenges to the safe and stable operation of district energy [3].

Digital technologies have evolved unprecedentedly in recent years, dramatically changing how devices communicate, monitor, analyze and display data [4]. These new technologies and the new business models and interactions they support continue to be incorporated into energy systems, bringing a wave of energy digitization across the globe. In the above context, integrating building energy systems has received increasing attention, which can effectively reduce energy operating costs and improve efficiency by utilizing advanced technologies such as smart meters, machine learning, and big data analytics to provide energy users with an economical and sustainable operating environment. Moreover, through intelligent scheduling of building energy systems, buildings can achieve an optimal trade-off between energy costs, carbon emissions and user comfort.

Meanwhile, the rapid development of energy conversion and storage devices such as cogeneration, heat pumps, gas boilers, thermal and electrical energy storage, and distributed energy solutions such as the integration of light, storage and charging has promoted the continuous integration of natural gas, oil, coal and electric power resources in the region, and the deepening of the multi-energy coupling and information interactions, which have gradually resulted in the formation of an integrated energy system at different levels. Although integrated energy systems in buildings can be a crucial option for the decarbonization of regional energy systems, the complexity of their building thermodynamic models, the large number of system uncertainty parameters, and the coupling of energy subsystems have brought many challenges to research and application [5].

Many scholars have recently studied integrated energy systems or similar systems for buildings. Literature [6] proposes a hybrid energy system that combines various energy generation, storage, and conversion technologies to effectively enhance grid integration of renewable energy sources with the goal of low cost and high efficiency. Literature [7] proposed a robust optimization method for regionally integrated energy systems considering the uncertainty of distributed energy stations, which divides the regionally integrated energy system into the upper electrical energy supply network and the lower distributed energy stations, effectively reducing the system operation instability caused by uncertainty factors and improves the system's anti-interference ability. Literature [8] proposes a low-carbon optimal scheduling method for integrated energy systems, which integrates the carbon emission model into the integrated energy system, realizes the complementarity of multiple energy sources and energy use substitution, and effectively solves the problem of high carbon emission. Literature [9] evaluated the feasibility of building integrated battery energy storage systems in the industry, quantifying the project's economic feasibility with the help of an iterative battery capacity algorithm to obtain the optimal battery capacity.

In integrated building energy systems, the synergistic configuration of renewable energy sources and energy storage devices enhances the flexibility of system scheduling but also brings characteristics such as uncertainty and complexity of constraints to the corresponding decision-making problems, which leads to difficulties

in optimization modelling and solving [10]. Meanwhile, traditional optimization algorithms can only converge to the optimal solution by assuming the convex characteristics of the system and usually require a considerable number of iterations. In contrast, as a general artificial intelligence technique, reinforcement learning algorithms can effectively cope with uncertainty by continuously acquiring knowledge through the interaction between the intelligence and the environment in which it is embedded and assisting the system in making decisions without relying entirely on the mathematical model. In existing work, some reinforcement learning-based approaches have been proposed for energy management in commercial buildings. Deep Reinforcement Learning (DRL) based approaches have been successfully applied to the building energy management problem to effectively cope with uncertainty and support the operation of integrated energy systems in buildings. However, in real scenarios, there are still many problems with the implementation of DRL. For example, considering the ample action space and state space, the performance of the intelligences is easily affected by the collected samples, which reduces the training speed and effectiveness of the models. Therefore, in this paper, based on the low-carbon energy management method for a single building, a multi-building intelligence co-optimization method is designed to improve the training speed of the building energy management system model.

The integrated energy system for intelligent buildings proposed in this paper is a user-oriented terminal multi-energy system, which has many types of energy devices inside, such as power generation devices such as photovoltaic panels, wind turbines, diesel generators, energy storage devices such as electricity, heat, air compression, energy conversion devices such as combined heat and power (CHP), electric heat pumps, gas boilers and other energy-using devices, such as lighting systems, shuttering systems, electric vehicles and other power-using devices. Since the operation of these devices can have considerable economic and environmental impacts on buildings, it is necessary to coordinate their scheduling. An integrated energy system for intelligent buildings can reduce the energy costs of smart buildings and promote the penetration of renewable energy on the user side while meeting the energy needs of end-users and the comfort of the service.

This paper builds upon existing research to address the low-carbon operation of integrated energy systems in buildings. The system architecture framework is illustrated in Fig 1. To overcome challenges such as the complexity of model construction, numerous uncertain parameters, spatiotemporal coupling of energy subsystems, and the time-consuming computational processes of existing methods, this paper proposes a novel integrated energy system for low-carbon buildings. A new mathematical model is developed for the low-carbon integrated energy system, which effectively handles system uncertainties without requiring explicit knowledge of the building's thermodynamic model. This approach enables real-time optimal capacity scheduling for a wide range of energy device combinations within the building.

## 2 Modelling of integrated energy systems in buildings

### 2.1 Structure of integrated energy systems in buildings

An integrated building energy system is designed to satisfy the electrical load (Electric Load, EL) and heat load demand (Heat Load, HL) of the occupants in a building through the synergistic action of different energy components. The structure of the integrated building energy system proposed in this paper is shown in Fig 2:

1) Heating, Ventilation, and Air Conditioning (HVAC): this system is used to maintain indoor comfort in buildings. It includes heating, ventilation, and air conditioning functions to meet the comfort needs of the occupants by controlling air temperature, humidity, and flow [11].

2) Renewable power generators are primarily solar photovoltaic (PV) systems. Solar PV systems power buildings by converting sunlight into electricity. Solar PV systems are usually installed on the roofs of buildings and use photovoltaic panels to convert solar energy into direct current (DC), which is then converted into alternating current (AC) by an inverter for use in the building [12].

 

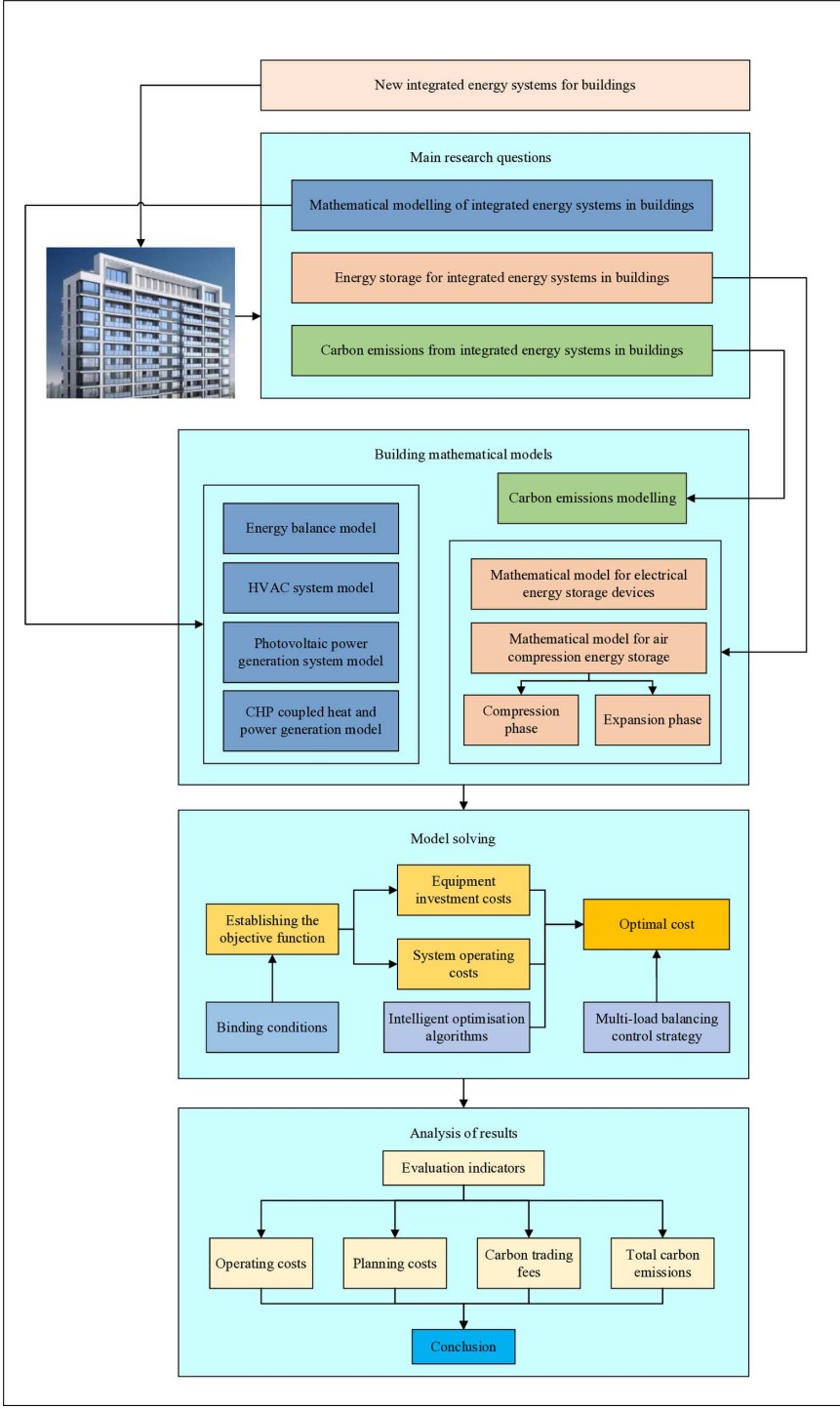

**Fig 1. Diagram of the overall framework of the study.**

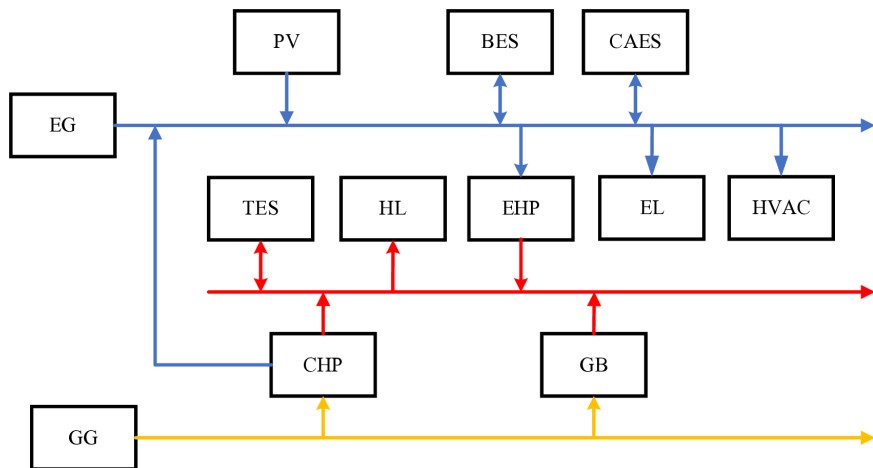

**Fig 2. Comprehensive building energy system structure.**

3) Energy storage devices include the Electrical Energy Storage System (EES) and Thermal Energy Storage System (TES). An electrical energy storage system consists of a mixture of Battery Energy Storage (BES) and Compressed Air Energy Storage (CAES) to store electricity to meet peak demand and release the stored electricity during low-demand periods [13]. Thermal energy storage systems, on the other hand, store thermal energy to supply heating to buildings [14].

4) Energy conversion devices include a Combined Heating and Power (CHP) system, an Electrical Heat Pump (EHP) and a Gas Boiler (GB). CHP systems can provide both heat and electricity, making efficient use of energy resources [15]; Electrical Heat Pumps drive heat pumps through electricity to absorb low-temperature heat energy and raise the temperature for heating or cooling; and Gas Boilers generate heat energy by burning gas to meet the heating and hot water needs of buildings.

The integrated energy system for buildings proposed in this paper can achieve a more flexible, efficient and sustainable power supply by combining a battery storage system and a compression energy storage system to meet the peak demand for electricity and to promote the application of clean energy and the sustainable development of the energy system to meet the energy needs of different types of buildings.

## 2.2 Modelling of integrated energy systems in buildings

The building energy storage system is a critical energy management solution, playing a pivotal role in storing and utilizing energy efficiently. Its structure primarily consists of key components such as energy storage units, energy converters, monitoring and control systems, energy management systems, and connection interfaces. The proposed structure of the building energy storage system in this study is illustrated in Fig 3. Designed to enhance building energy utilization efficiency, reduce energy costs, and improve energy security, this system provides tailored support and solutions for sustainable energy management in buildings.

Integrated building energy systems need to meet the energy balance equation during operation, which is expressed as follows:

$$PB(t) + P_{CHP}(t) + PV(t) + P_{ED}(t) = EL(t) + P_{EHP}(t) + P_{EC}(t) + P_{HC}(t) \tag{1}$$

$$Q_{CHP}(t) + Q_{EHP}(t) + Q_{GB}(t) + Q_{TD}(t) = HL(t) + Q_{TC}(t) \tag{2}$$

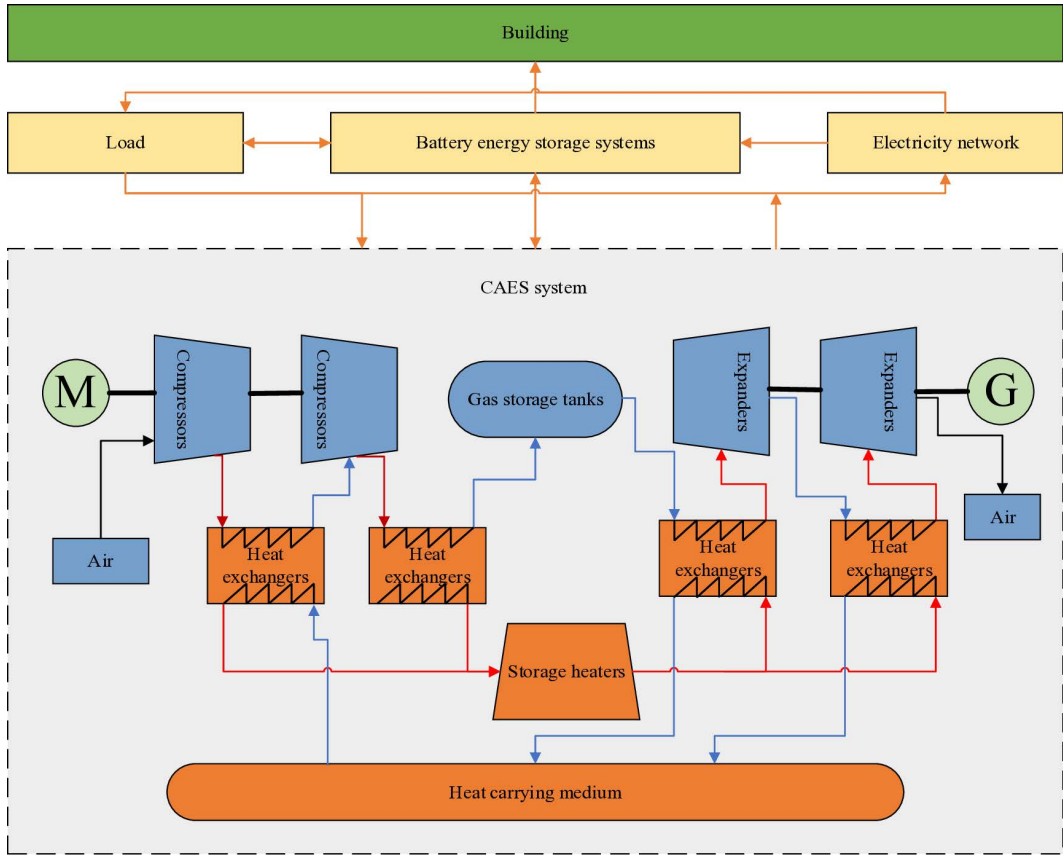

**Fig 3. Structure of building energy storage system.**

$PB(t)$ denotes the amount of electricity purchased by the building from the grid at time t;

$PV(t)$ denotes the distributed PV output power at moment t;

$EL(t)$ denotes the electrical load of the building at time t;

$HL(t)$ denotes the heat load of the building at time t;

$P_{HC}(t)$ denotes the electrical power used by the HVAC system;

$P_{CHP}(t)$ denotes the power output of CHP at time t;

$Q_{CHP}(t)$ denotes the heat output of CHP at time t;

$P_{EHP}(t)$ denotes the input power of the EHP;

$Q_{EHP}(t)$ denotes the heat output of the EHP;

$Q_{GB}(t)$ denotes the heat output of GB;

$P_{EC}(t)$ denotes the charging power of the electric energy storage EES;

$P_{ED}(t)$ denotes the discharge power of the electrical energy storage EES;

$Q_{TC}(t)$ denotes the absorbed power at the moment t of the thermal energy storage unit TES;

$Q_{TD}(t)$ denotes the output power of the thermal energy storage unit TES at time t;

**2.2.1 Energy storage devices.** Energy storage devices can redistribute off-peak and peak loads and absorb free renewable energy for future use when energy prices peak [16]. The mathematical model of an electrical energy storage device EES can be expressed as:

$$0 \ll P_{EC}(t) \ll P_{ES}^{MAX} \tag{4}$$

$$-P_{ES}^{MAX} \ll P_{ED}(t) \ll 0 \tag{5}$$

$$0 \ll E_{ES}(t) \le E_{ES}^{MAX} \tag{6}$$

$$E_{ES}(t+1) = E_{ES}(t) + P_{EC}(t) \Delta t \eta_{ES} + P_{ED}(t) \Delta t / \eta_{ES} \tag{7}$$

**2.2.2 Compressed air energy storage.** Compressed air energy storage (CAES) uses excess electricity to drive the mechanism to work [17]. The energy storage process mainly uses electric power to drive the compressor to convert the wind energy into high-pressure air in the storage tank to achieve energy storage. In contrast, the energy release process feeds the high-pressure air into the combustion chamber to drive the expander to start working and to convert the air pressure energy into electricity. The mathematical model of the compression phase, as well as the expansion phase of the CAES system, is shown below:

Compression phase:

$$P_{CAES}^{ch}(t) = \sum_{i=1}^{N_{com}} \frac{\mu R_g}{\mu - 1} q_{com}(t) \left( T_{com,out}^i - T_{com,in}^i \right) \tag{8}$$

$$T_{com,out}^i = T_{com,in}^i \left[ 1 + \left( \pi_{com,i}^{\frac{\mu-1}{\mu}} - 1 \right) / \eta_{com} \right] \tag{9}$$

$\eta_{com}$ denotes the isentropic efficiency of the compressor;
$P_{CAES}^{ch}(t)$ denotes the charging power of the system;
$N_{com}$ denotes compressor factor;
$R_g$ denotes the gas constant;
$q_{com}(t)$ denotes mass flow rate;
$T_{com,in}^i$ denotes inlet temperature;
$T_{com,out}^i$ denotes outlet temperature;
$\pi_{com,i}$ denotes the rated compression ratio;
Expansion phase:

$$P_{CAES}^{dis}(t) = \sum_{i=1}^{N_{exp}} \frac{\mu R_g}{\mu - 1} q_{exp}(t) \left( T_{exp,in}^i - T_{exp,out}^i \right) \tag{10}$$

$$T_{exp,out}^i = T_{exp,in}^i \left[ 1 - \left( 1 - \pi_{exp,i}^{\frac{\mu-1}{\mu}} \right) / \eta_{exp} \right] \tag{11}$$

$\eta_{exp}$ denotes the isentropic efficiency of the expander;
$P_{CAES}^{dis}(t)$ denotes system discharge power;

$N_{exp}$ denotes the number of expander stages;

$q_{exp}(t)$ denotes mass flow rate;

$T^i_{exp,in}$ denotes inlet temperature;

$T^i_{exp,out}$ denotes outlet temperature;

$\pi_{exp,i}$ denotes the nominal expansion ratio;

**2.2.3 HVAC system.** HVAC systems are usually followed in residential and commercial buildings to maintain the indoor temperature within the desired comfort range. Therefore, operating an HVAC system converts electricity to thermal comfort [18]. The mathematical model of an HVAC system can be expressed as:

$$0 \ll P_{HC}(t) \ll P_{HC}^{MAX} \tag{12}$$

$$H_{HVAC}^{MIN} \ll H_{IN}(t) \ll H_{HVAC}^{MAX} \tag{13}$$

$$H_{IN}(t+1) = \frac{H_{IN}(t) - (H_{IN}(t) - H_{OUT}(t) + \eta_{HVAC}R_{HVAC}P_{HC}(t))\,\Delta t}{C_{HVAC}R_{HVAC}} \tag{14}$$

$H_{IN}$ denotes room temperature;

$H_{OUT}$ denotes outdoor temperature;

$P_{HC}$ denotes HVAC system power;

$\eta_{HVAC}$ denotes HVAV system efficiency;

$C_{HVAC}$ denotes building thermal capacity;

$R_{HVAC}$ indicates thermal resistance;

**2.2.4 Modelling of photovoltaic power generation systems.** The combination of PV power systems and buildings has become the main application of PV systems in cities [19]. In this paper, the building system places PV arrays on the roof of the building and PV modules on the exterior walls. The PV power generation system model can be expressed as:

$$P_{PV}^{MAX}(t) = S_{PV}\eta_{PV}\eta_{LOSS}I_{PV}(t) \tag{15}$$

$$\eta_{LOSS} = 1 - 0.0045\,(T_{OUT}(t) - 25) \tag{16}$$

$$0 \leq P_{PV}(t) \leq P_{PV}^{MAX}(t) \tag{17}$$

$P_{PV}(t)$ a denotes the actual output of the PV system at time t;

$P_{PV}^{MAX}(t)$ denotes the maximum output power that the photovoltaic power generation system can produce at the moment t;

$S_{PV}$ denotes the area of the PV panel;

$\eta_{PV}$ denotes the power generation rate of the photovoltaic system;

$\eta_{LOSS}$ denotes the power of the photovoltaic system after power loss due to temperature increase;

$I_{PV}(t)$ denotes the amount of solar irradiation captured per unit area by the PV system at time t;

$T_{OUT}(t)$ represents the ambient temperature at time t;

**2.2.5 Carbon emission modelling.** Carbon emission modelling is a tool used to assess and predict carbon emissions from different activities or systems. It is based on various data and indicators to estimate and analyze the carbon

emissions of a particular activity or system through mathematical algorithms and simulation methods [20]. The time-measurement model of carbon emissions from upstream power purchases is shown below:

$$\begin{cases} M_E(t) = \varepsilon_E(t) P_E(t) \\ \varepsilon_E(t) = R_S(t) \psi_S \end{cases} \tag{18}$$

$M_E(t)$ denotes the power purchase equivalent carbon emissions of the parent grid at moment t;

$\varepsilon_E(t)$ denotes carbon emission measurement factor;

$P_E(t)$ denotes the interaction rate with the building;

$R_S(t)$ denotes the unit share of electricity generation of the power unit at time t;

$\psi_S$ denotes carbon emission measurement factor per unit of electricity generation;

### 2.3 Energy conversion equipment

**2.3.1 Cogeneration systems.** Cogeneration is an input-multiple-output converter typically characterized by high energy efficiency compared to independent power and heat sources [21]. The mathematical model of CHP's thermoelectric coupled generation can be expressed as follows:

$$P_{CHP}(t) = \eta_{CHPE} G_{CHP}(t) \tag{19}$$

$$Q_{CHP}(t) = \eta_{CHPQ} G_{CHP}(t) \tag{20}$$

$$0 \ll P_{CHP}(t) \ll P_{CPH}^{MAX} \tag{21}$$

$$0 \ll Q_{CHP}(t) \ll Q_{CHP}^{MAX} \tag{22}$$

**2.3.2 Electric heat pumps.** An electric heat pump is a device that uses electrical energy to convert and transfer heat energy [22]. It achieves the transfer of heat energy by absorbing heat from a low-temperature heat source and releasing it to a high-temperature heat source through the principle of cyclic operation.

$$P_{EHP}(t) = \eta_{EHP} P_{EHP}(t) \tag{23}$$

$$0 \ll Q_{EHP}(t) \le Q_{EHP}^{MAX} \tag{24}$$

**2.3.3 Gas boilers.** A gas boiler is a device that uses natural gas or liquefied petroleum gas (LPG) for combustion to generate heat energy [23]. It generates high-temperature combustion products by burning the fuel to transfer heat energy to water or other media to achieve the function of heating, warming or hot water.

$$Q_{GB}(t) = \eta_{GB} G_{GB}(t) \tag{25}$$

$$0 \ll Q_{GB}(t) \ll Q_{GB}^{MAX} \tag{26}$$

**2.3.4 Gas water heaters.** A gas water heater is a device that uses a fuel such as natural gas or liquefied petroleum gas (LPG) for combustion to provide hot water [24]. It generates heat energy by burning the fuel, which is transferred to the water stream to heat it to the desired temperature.

$$H^i_{GH}(t) = G^i_{GH}(t)\,\eta_{GH} \tag{27}$$

$H^i_{GH}(t)$ denotes the gas energy consumed by the gas water heater i at time t;
$G^i_{GH}$ denotes the amount of heat supplied by gas water heater i at time t;
$\eta_{GH}$ denotes the efficiency of the gas water heater;

**2.3.5 Induction cookers.** Buildings are equipped with induction cookers and gas cookers, and users can choose between them according to the price of electricity and gas to realize the mutual substitution of electricity and gas on the user side.

$$\begin{cases} P^i_{EG}(t)\,\eta_{EG} = G^i_{GE}(t)\,\eta_{GE} \\ P^i_{EG}(t) = G^i_{GE}(t)\,\eta_{GE}/\eta_{EG} \\ G^i_{GE}(t) = P^i_{EG}(t)\,\eta_{EG}/\eta_{GE} \end{cases} \tag{28}$$

$P^i_{EG}(t)$ denotes the electric power of the induction cooker i at time t;
$G^i_{GE}$ denotes the gas power of gas cooker i at time t;
$\eta_{EG}$ denotes the efficiency of the induction cooker;
$\eta_{GE}$ denotes the efficiency of the gas cooker;

## 2.4 Objective function

In this paper, the equivalent annual value method is used to convert the investment cost into the equivalent annual investment cost over the life cycle of the equipment, and the planning is carried out in the initial year with the optimization objective of minimizing the equivalent investment cost and operating cost in a single year. The objective function of the planning model can be expressed as:

$$\min C = \min\left(C_{INV} + C_{OPE}\right) \tag{29}$$

$C_{INV}$ denotes annualized equipment investment costs;
$C_{OPE}$ denotes system operating costs;

$$C_{INV} = \sum_{i=1}^{N} n_i C_i \frac{r(1+r)^{T_i}}{(1+r)^{T_i} - 1} \tag{30}$$

$N$ denotes the total number of devices;
$n_i$ denotes the number of units configured for device i;
$C_i$ denotes the cost of device i configuration;
$T_i$ denotes the life cycle of equipment i;
$r$ denotes discount rate;

$$C_{OPE} = 365 \sum_{KD} \theta_{KD}\left(C_{OM} + C_{ENERY} + C_{UNC}\right) + C_{CO_2} \tag{31}$$

$$C_{OM} = \sum_{i=1}^{N} \sum_{t=1}^{24} c_i^{om} P_i \Delta t \tag{32}$$

$$C_{ENERY} = \sum_{t=1}^{24} \left( c_t^E P_t^E + c_t^G + G_t^G \right) \Delta t \tag{33}$$

$$C_{UNC} = \sum_{i=1}^{24} \varepsilon_{UNC} \left( H_{IN}(t) - H_{SET} \right) \Delta t \tag{34}$$

$C_{OM}$ denotes the cost of operating and maintaining the equipment of the building system;

$C_{ENERY}$ denotes the cost of purchased energy;

$C_{UNC}$ denotes penalty charge for uncomfortable indoor temperatures;

$KD$ denotes typical day type;

$\theta_{KD}$ denotes the proportion of typical days of a certain type in the year;

$c_i^{om}$ denotes the operation and maintenance cost per unit of power output of equipment i;

$c_t^E$ denotes the purchase price of electricity at time t;

$c_t^G$ denotes the purchase price of gas energy at time t;

$G_t^G$ denotes the purchased gas power at time t;

$\varepsilon_{UNC}$ denotes the indoor temperature discomfort cost factor;

$H_{SET}$ denotes the indoor temperature setting;

## 2.5 Multiple load balancing control strategy

In this paper, we propose a multi-load balancing control strategy (MLBS), whereby when the external energy tariffs change, the building adjusts its energy load to some extent according to the price changes.

$$\begin{bmatrix} Ld_e(t) \\ Ld_g(t) \\ Ld_h(t) \end{bmatrix} = \begin{bmatrix} Ld_e^0(t) \\ Ld_g^0(t) \\ Ld_h^0(t) \end{bmatrix} + \begin{bmatrix} \Delta Ld_e(t) \\ \Delta Ld_g(t) \\ \Delta Ld_h(t) \end{bmatrix} + \begin{bmatrix} V_{Ech}(t) + V_{Ecg}(t) - V_e(t) \\ V_{Gch}(t) + V_{Gcg}(t) - V_g(t) \\ - V_h(t) \end{bmatrix} \tag{35}$$

$Ld_e(t)$ denotes the electrical load of the building after equalization control at time t;

$Ld_g(t)$ denotes the air load of the building after equalization control at time t;

$Ld_h(t)$ $Ld_e^0(t)$ denotes the heat load of the building after equalization control at time t;

denotes the electrical load of the building before equalization control at time t;

$Ld_g^0(t)$ denotes the air load of the building before equalization control at time t;

$Ld_h^0(t)$ denotes the heat load of the building before equalization control at time t;

$\Delta Ld_e(t)$ denotes the amount of change in the adjustable portion of the electrical load due to equalization control;

$\Delta Ld_g(t)$ denotes the amount of change in the adjustable portion of the gas load due to equalization control;

$\Delta Ld_h(t)$ denotes the amount of change in the adjustable portion of the heat load due to equalization control;

$V_{Ech}(t)$ denotes the increase in electrical load due to electrical heat exchange;

$V_{Gch}(t)$ denotes the increase in gas load due to gas heat exchange;

$V_h(t)$ denotes the reduction in heat load due to energy exchange;

$V_{Ecg}(t)$ denotes the increase in electrical energy due to electrical substitution;

 

$V_{Gcg}(t)$ denotes the increase in gas energy due to electrical substitution;

$V_e(t)$ denotes the reduction in electrical energy due to electrical substitution;

$V_g(t)$ denotes the reduction in gas energy due to electrical substitution;

When external energy prices change, the price response occurs in the commercial area of the building, with varying degrees of increase, decrease, and levelling off of electricity, gas, and heat loads. Transferable Electricity or Gas Load: The transferable electricity or gas load for each period is within a certain range, and the total amount of electricity or gas load remains unchanged throughout the day. Transferable thermal load: The indoor temperature changes within a reasonable range in a short period to a small degree, and the user's thermal comfort will not be greatly affected. The constraints are shown in the following equation:

$$\begin{cases} -\Delta Ld_e^{\max}(t) \le \Delta Ld_e(t) \le \Delta Ld_e^{\max}(t) \\ -\Delta Ld_g^{\max}(t) \le \Delta Ld_g(t) \le \Delta Ld_g^{\max}(t) \\ \sum_{t=1}^{T} \Delta Ld_e(t) = 0 \\ \sum_{t=1}^{T} \Delta Ld_g(t) = 0 \end{cases} \tag{36}$$

$\Delta Ld_e^{\max}(t)$ denotes the maximum electrical load that can be transferred at time t;

$\Delta Ld_g^{\max}(t)$ denotes the maximum air load that can be transferred at time t;

$$\begin{cases} Ld_h(t) = H_{IN}(t) + H_{AC}(t) \\ Ld_h^0(t) = H_{IN}^0(t) + H_{AC}^0(t) \\ \Delta Ld_h(t) = Ld_h(t) - Ld_h^0(t) \end{cases} \tag{37}$$

$H_{IN}(t)$ a denotes an indoor heat source in the building;

$H_{AC}(t)$ a denotes the heat production of the air conditioner at time t;

The incremental amount of electric and gas energy demand due to electric heat exchange and electrical exchange, the amount of energy change due to gas-electricity substitution, and the amount of heat energy reduction due to energy substitution are shown in the following equations:

$$\begin{cases} V_{Ech}(t) = P_{AC}(t) \\ V_{Gch}(t) = G_{GH}(t) \\ V_{Ecg}(t) = P_{EG}(t) \\ V_{Gcg}(t) = G_{GE}(t) \\ V_e(t) = G_{GE}(t)\,\eta_{GE}/\eta_{EG} \\ V_g(t) = P_{EG}(t)\,\eta_{EG}/\eta_{GE} \\ V_h(t) = H_{AC}(t) + H_{GH}(t) \end{cases} \tag{38}$$

## 3 Improved Tuna Swarm Algorithm

Tuna Swarm Optimization (TSO) [25] is a heuristic optimization algorithm based on the behaviour of tuna in nature. The algorithm simulates the behavioural strategies of tuna during foraging and is used to solve various optimization issues. The basic idea of the TSO algorithm is to perform global search and optimization by learning and simulating how tuna move. Tuna are characterized by fast, agile and cooperative behaviour during foraging to find food sources efficiently. The algorithm introduces these characteristics into the optimization problem to improve the search efficiency and solution quality.

However, the tuna swarm optimization algorithm in the early convergence speed is slow and quickly falls into the local optimum and other shortcomings. The introduction of the Circle chaotic mapping initialization strategy, Levy flight (Levy flight) strategy, and adaptive inertia weight factor to improve the tuna optimization algorithm, the formation of improved Tuna Swarm optimization algorithm (Improved Tuna Swarm Optimization), the specific improvement strategy is shown in Fig 4.

(1) Use Circle chaotic mapping to initialize the tuna population so that more randomness is introduced into the initial selection of the tuna population, thus increasing the diversity of individuals in the population and better covering the problem's solution space [26]. The specific formula is as follows:

$$x_{i+1} = \text{mod}\left(x_i + 0.2 - \left(\frac{0.5}{2\pi}\right)\sin(2\pi \cdot x_i), 1\right)$$

(39)

Where mod is the residual function and $x_{i+1}$ denotes the value of the i + 1st mapping.

(2) Improve the tuna spiral foraging with Levy flight. When the optimal individual in the tuna group does not find food, the other tuna in the group update their position according to Levy flight, which helps the TSO algorithm to improve the

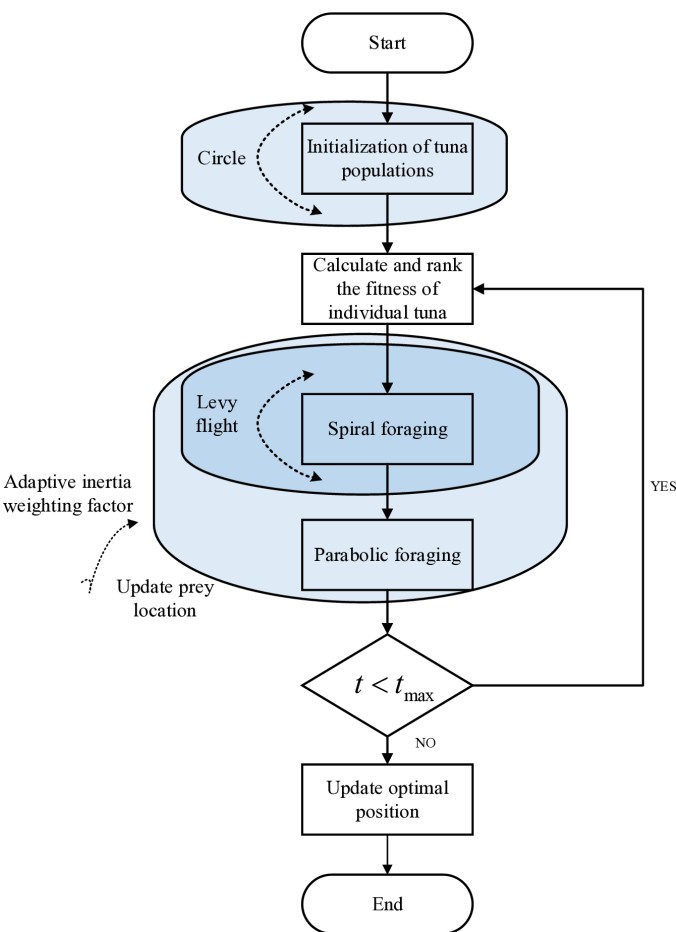

**Fig 4. Flowchart of the ITSO algorithm.**

spatial searching ability and the ability to jump out of the local optimum, which will help to find the global optimum [27]. The specific formula is as follows:

$$X_i^{t+1} = \begin{cases} k_1 \times X_i^t \times Levy(D) + k_2 \times X_i^t, i = 1 \\ k_1 \times X_i^t \times Levy(D) + k_2 \times X_i^t, 1 < i < it\,max \end{cases} \tag{40}$$

$$Levy(x) = 0.01 \cdot \frac{u}{|v|^{\frac{1}{2}}} \tag{41}$$

$$u \sim N\left(0, \sigma_u^2\right), v \sim N\left(0, \sigma_v^2\right) \tag{42}$$

$$\sigma_u = \left\{ \frac{\Gamma(1+\lambda)\sin\left(\frac{\pi\lambda}{2}\right)}{\lambda \times \Gamma(\frac{1+\lambda}{2}) \times 2^{\frac{\lambda-1}{2}}} \right\}^{\frac{1}{\lambda}}, \sigma_v = 1 \tag{43}$$

$$\Gamma = \int_0^{+\infty} e^{-t} t^{x-1} dt \tag{44}$$

Among them, $k_1$ and $k_2$ control the weight coefficients of the individual to Levy flight individual and the previous individual moving trend; $\lambda$ value is generally [1,3]; $\sigma_u \sigma_v$ is the normal distribution function, u and v are in line with the normal distribution of $\sigma_u \sigma_v$, $N$ is a normal distribution; $D$ is the dimensionality of the position vector;

(3) Introducing the inertia weighting factor can allow the tuna optimization algorithm to use a relatively hearse amount of time for the global search; when the inertia weighting factor is small, the algorithm uses a relatively large amount of time for the local search, which can be fine-tuned to find the optimal solution [28]. The specific formula is expressed as follows:

$$w = \sin\left(\frac{\pi t}{2t_{max}} + \pi\right) + 1 \tag{45}$$

$$x_i'^{t+1} = w \cdot x_i^{t+1} \tag{46}$$

Where w is the adaptive inertia weighting factor;

The flowchart of the ITSO algorithm is depicted in Fig 4. The ITSO algorithm employs Circle chaotic mapping to initialize the population, enhancing the diversity and richness of the initial solutions. To improve the algorithm's performance, the search characteristics of Levy flight are utilized to enhance the exploration capability during spiral foraging, thereby reducing the likelihood of the algorithm becoming trapped in local optima and accelerating the convergence to the global optimum. Additionally, an adaptive weighting method is introduced to refine the local exploitation capability of the algorithm. This involves re-updating the neighborhood of the prey location to identify more optimal solutions, ensuring a balance between exploration and exploitation throughout the optimization process.

## 4 Analysis of results

The building in this paper is a building complex because different functional areas have different energy demand characteristics. The energy consumption patterns and peak and valley load characteristics of the three regional functional areas of Office Building, Gourmet City, and Apartment Building are apparent differences. Analyzing these functional areas makes it possible to analyze their energy usage better and thus target energy management strategies. This building complex is located in Huai'an, Jiangsu Province, China. Based on the actual energy demand data of a building complex, the Office Building, Gourmet City, and Apartment Building within the complex were selected as typical functional areas for analysis.

While realizing the demand for energy supply, it is more important to consider its economic and environmental sustainability. Analyzing the comprehensive influencing factors, such as the various regions of the building business, can provide a scientific basis for the low-carbon planning of the building and formulate a reasonable energy regulation strategy and equipment configuration program. Therefore, through the model of building energy equipment, the installed capacity of the building energy centre equipment, and the equipment planning capacity settings, the optimal equipment can be achieved through the careful consideration of these factors to meet the demand for energy supply in the commercial area and optimize the efficiency of energy use. Such as Tables 1 and 2 planning settings, according to the equipment capacity, life, investment costs, maintenance costs, thermal efficiency, electrical efficiency charging and discharging power and efficiency, and other indicators set to achieve the demand for energy supply to the commercial areas of the building, in order to achieve the goal of reducing carbon emissions and achieve sustainable development.

Low carbon planning for buildings involves the configuration of office and building energy equipment to meet the demand for energy supply to commercial areas of the building. By configuring the models and number of units of energy equipment in office buildings and buildings, the demand for energy supply to the commercial areas of the building is realized. The load data of each functional area of the building comes from the measured data of a power distribution building. The daily load data of the typical scenarios selected from the three areas of Office Building, Gourmet City, and Apartment Building are plotted in Fig 5. The purchase price of electricity and natural gas is taken as a reference for the yearly data of a city in Jiangsu Province, north of Jiangsu Province. The installation of photovoltaic panels is placed on the roof of the building, as the photovoltaic panels are installed. The PV panels are installed on the roof of the building and the façade with the best illumination.

For the low carbon planning model constructed in this paper considering integrated buildings, this paper is divided into 2 cases for comparison.

**Table 1. Integrated building equipment planning parameters.**

| Device | Capacity (kWh) | Lifespan (years) | Operating cost (¥million) | Maintenance cost (¥/kW/h) | Thermal efficiency | Generating efficiency |
|--------|---------------|------------------|--------------------------|---------------------------|-------------------|----------------------|
| PV | 0.53 | 14 | 0.265 | 0.068 | \ | \ |
| EHP | 110 | 22 | 39 | 0.036 | 0.97 | \ |
| GB | 170 | 27 | 182 | 0.027 | 0.96 | \ |
| CHP | 220 | 21 | 110 | 0.015 | 0.3 | 0.6 |

**Table 2. Planning parameters for energy storage equipment.**

| Device | Capacity (kWh) | Charge/discharge power (kW) | Charge/discharge efficiency | Lifespan (years) | Operating cost (¥million) | Maintenance cost (¥/kW/h) |
|--------|---------------|----------------------------|----------------------------|------------------|---------------------------|---------------------------|
| BES | 500 | 110/110 | 0.87/0.87 | 18 | 47 | 0.015 |
| CAES | 2237 | 420/420 | 0.68/0.68 | 27 | 128 | 0.005 |

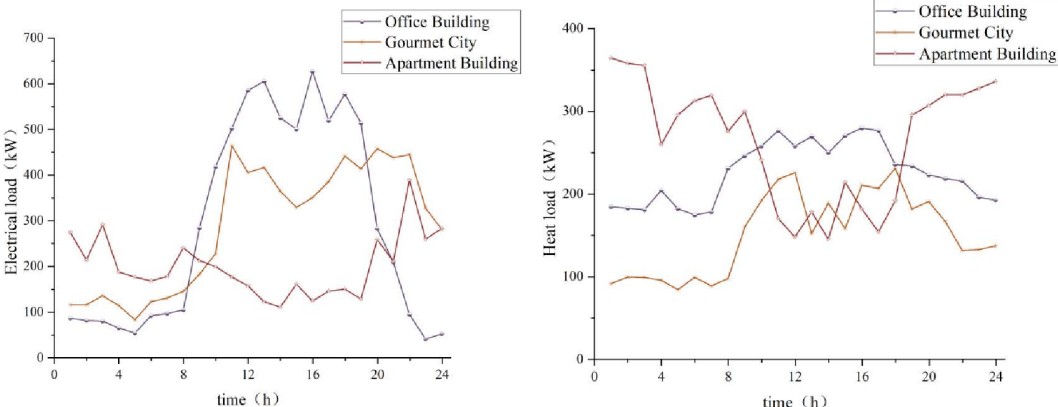

**Fig 5. Electricity/heat loads in the commercial area of the building on a typical scenario day.**

(1) Case 1: Using a fixed carbon emission strategy for optimization without considering the Multi-load balancing control strategy (MLBS).

(2) Case 2: Multi-load balancing control strategy (MLBS) and optimize the carbon emissions using the low-carbon economic dispatch model.

As seen in Fig 6, the electric and thermal loads have different degrees of change in each period after the effect of MLBS and the low carbon economy dispatching strategy. 0:00~8:00 electric and thermal loads have lower utilization rates, the effect of MLBS and low carbon economy dispatching strategy is small, and there are different degrees of slight increase in Case2 compared to Case1; 8:00~17:00 electric and thermal loads have higher utilization rates, the effect of MLBS and low carbon economy dispatching strategy is small; 8:00~17:00 electric and thermal loads have higher utilization rates, the effect of MLBS and low carbon economy dispatching strategy is small; 8:00~17:00 electric and thermal loads have higher utilization rates, the effect of MLBS and low carbon economy dispatching strategy is small. The utilization rate of electric and thermal loads is higher, and the effects of MLBS and low carbon economy scheduling strategy are more pronounced. The

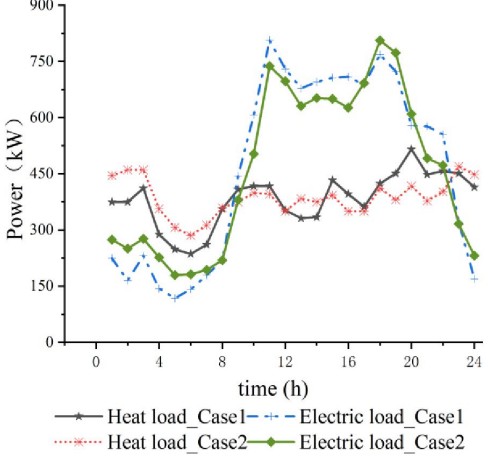

**Fig 6. Electrical and thermal load curves before and after optimization.**

electric load before and after regulation decreases, and the thermal load increases; 17:00~23:00, the load power gradually decreases, the electric load before and after regulation increases and then decreases, and the thermal load decreases.

Further, this paper analyzes the results of low-carbon planning for buildings, taking Case1 and Case2 as examples, and evaluates the system in terms of quarterly planning costs, operating costs, carbon trading costs, and total carbon emissions, respectively. Low-carbon building planning can reduce energy and maintenance costs and improve by optimizing the planning and operating costs. Companies can save costs and increase profits while reducing carbon emissions. Through Fig 7, it can be seen that after using MLBS and the effect of low carbon economic scheduling strategy, the lowest values of planning cost, operation cost, carbon transaction cost, and total carbon emission in the four quarters of Case1 before optimization are 120.88 million, 400.12 tonnes, 9.09million, and 147.27 million respectively, which are significantly higher than the lowest values of planning cost, operation cost, carbon transaction cost, and total carbon emission in Case2 after optimization, which are 110.37 million, 204.28 ton, 8.15 million, and 137.27 million in the four quarters; it can be seen that the improvement measures of MLBS and low-carbon economic dispatch strategy proposed in this paper can not only improve the economy but also improve the system to a certain extent. It can also improve the carbon emission level of the system to some extent, realizing the double improvement of economic efficiency and carbon emission.

From Fig 8, as the carbon trading price increases, it increases the planning capacity of PV, TES, BES_CAES, and GB and decreases the planning capacity of EHP and CHP. This is consistent with the trend of change from not adopting the strategy mechanism to using the MLBS strategy and from adopting the fixed carbon incentive and penalty mechanism to the low carbon economy scheduling strategy because the increase of the low carbon economy scheduling strategy is essentially also the increase of the carbon trading intensity. From Fig 8, as the carbon penalty price base increases,

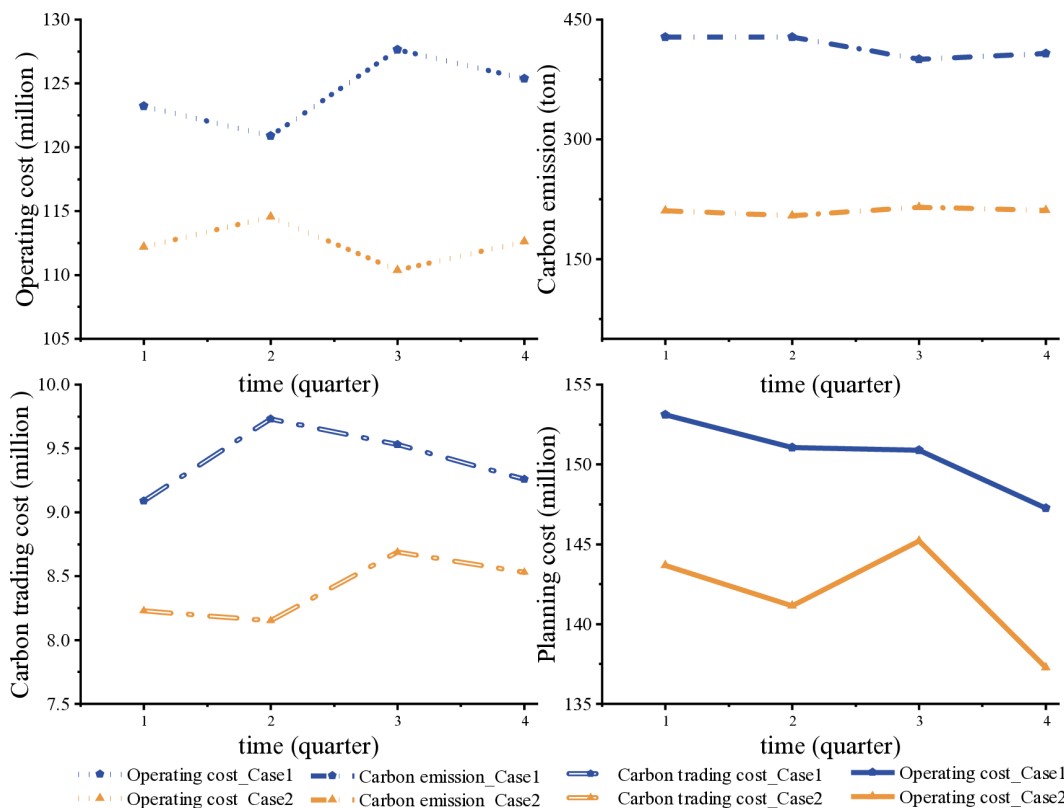

**Fig 7. Low carbon planning case study.**

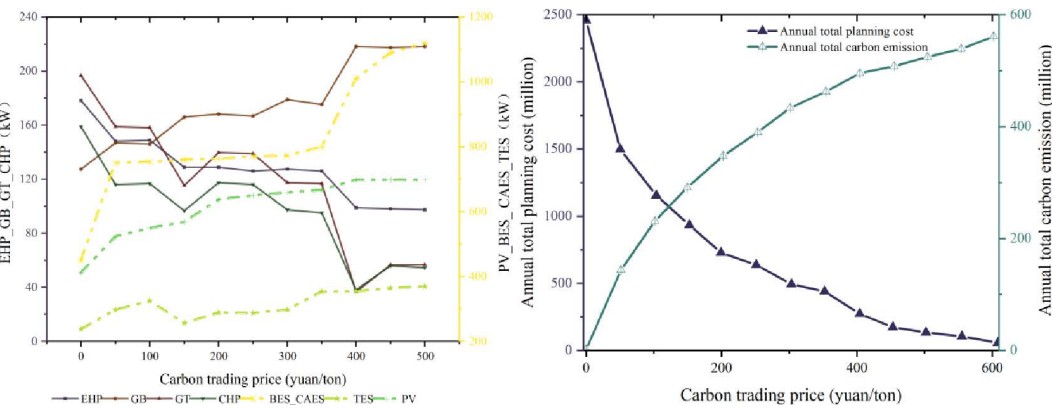

**Fig 8. Analysis of results of carbon trading mechanisms.**

the total annual planning cost increases while the carbon emissions decrease. From not considering the carbon penalty mechanism to considering the carbon penalty mechanism, the annual total planning cost and the total annual carbon emissions have the most apparent changes in magnitude. As the carbon trading price base grows to a certain magnitude, the annual total planning cost and the total annual carbon emissions are unchanged, showing a saturated state. As the carbon trading price and carbon penalty price increase, energy equipment's planning capacity and operation strategy change to reduce carbon emissions, improve energy efficiency and reduce total planning costs. These results provide a basis for effectively implementing low-carbon economic dispatch strategies.

To further highlight the innovation of the proposed system, as shown in Table 3, the load monitoring dynamic adjustment exhibits the lowest cost but requires longer computation time. The TSO-MLBS energy scheduling strategy adopted in this paper features the shortest computation time, lower energy scheduling cost, and optimal comprehensive performance.TSO-MLBS

## 5 Conclusion

The novel structure of the integrated building energy system proposed in this study realizes the flexible scheduling capability of the energy system by combining battery storage and compression storage, which provides a feasible solution for the energy demand of different building types. The innovatively proposed Multi-load balancing control strategy (MLBS) further reduces the quarterly planning cost, operating cost, carbon trading cost and total carbon emission of the building energy system and significantly improves economic cost and carbon emission level.

(1) The building energy system under the novel architecture proposed in this paper realizes a more flexible, efficient and sustainable power supply by flexibly configuring battery storage and compression storage elements to meet the demand during the peak electricity consumption period and to promote the application of clean energy and the

**Table 3. Comparison of the effect of energy scheduling strategies.**

| Energy Scheduling Strategy | Energy Dispatch Costs(¥) | Computation Time(s) |
|---|---|---|
| TSO-MLBS | 1033 | 1.32 |
| Load Balancing with Load Prediction [29] | 1159 | 2.56 |
| Load Balancing with Load Monitoring [30] | 1021 | 6.19 |
| Load Balancing with Feedback Control [31] | 1432 | 5.91 |
| Load Balancing with Fault-aware Adjustment [32] | 1335 | 9.35 |

sustainable development of the energy system. This system can flexibly configure battery energy storage and compression energy storage elements according to the characteristics and scale of its energy demand, providing a feasible solution to meet the energy demand of different buildings.

(2) Optimize energy scheduling through a multi-load balanced control strategy and construct a low-carbon economic scheduling model using the ITSO algorithm to achieve supply-demand balance and minimize costs. Through the effective combination of this system and MLBS, the system is optimized for different periods, load demands and energy prices so that the system can find the optimal balance between economic costs and carbon emissions, effectively improving the carbon emission level of the system, and reducing the carbon transaction costs and total carbon emissions.

## Author contributions

**Conceptualization:** Yefan Shu.

**Data curation:** Wenjie Wang.

**Formal analysis:** Yefan Shu, Zhengyu Wang, Chenliang Ji.

**Funding acquisition:** Jie Ji.

**Investigation:** Zhengyu Wang, Chenliang Ji, Hui Huang.

**Methodology:** Wenjie Wang, Hui Huang.

**Project administration:** Wenjie Wang, Hui Huang, Rundong Ji.

**Resources:** Rundong Ji, Mengxiong Zhou, Jie Ji.

**Software:** Mengxiong Zhou.

**Supervision:** Mengxiong Zhou.

**Validation:** Yaodong Wang.

**Visualization:** Yaodong Wang.

**Writing – original draft:** Yaodong Wang.

**Writing – review & editing:** Yaodong Wang.

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
