## [Decision Letter · Decision Letter 0]

Dear Dr. ji,

Thank you for submitting your manuscript to PLOS ONE. After careful consideration, we feel that it has merit but does not fully meet PLOS ONE’s publication criteria as it currently stands. Therefore, we invite you to submit a revised version of the manuscript that addresses the points raised during the review process.

We look forward to receiving your revised manuscript.

Kind regards,

Joy Nondy, Ph. D.

Academic Editor

PLOS ONE

3. In the online submission form, you indicated that [The datasets used and/or analysed during the current study available from the corresponding author (jijie@hyit.edu.cn) on reasonable request].

Reviewers' comments:

Reviewer's Responses to Questions

**Comments to the Author**

1. Is the manuscript technically sound, and do the data support the conclusions?

Reviewer #1: No

Reviewer #2: Yes

2. Has the statistical analysis been performed appropriately and rigorously?

Reviewer #1: N/A

Reviewer #2: Yes

3. Have the authors made all data underlying the findings in their manuscript fully available?

Reviewer #1: No

Reviewer #2: Yes

4. Is the manuscript presented in an intelligible fashion and written in standard English?

Reviewer #1: No

Reviewer #2: Yes

Reviewer #1: 1. The literature review should be totally reorganized to further clarify the research gap and the main contributions of this paper.

2. CAES and CHP are usually considered as a large equipment. It seems not practical to use them with a single building, but suitable for a park. Please further indicate the necessity or provide the evidence of the real project.

3. The meanings of different energy flow lines in Fig.2 should be indicated.

4. Eq(35) to realize the energy balancing or load control should be further explained.

5. The introduction of TSO to the proposed problem should be further explained.

6. the planning results in table 1 and 2 do not make sense. The capacity in Table 1 is not usually considered, while capacity in kW is considered. The capacity for BES and CAES is too large for a building which also occupy a large amount of land, increasing the capital cost.

7. comparisons with other existing methods should be further considered.

Reviewer #2: With the escalating global energy demand and urgent need for carbon reduction, this paper proposes a Multi-load Balancing Control Strategy (MLBS) based on an Improved Tuna Swarm Optimization (ITSO) algorithm for a novel low-carbon integrated energy system in buildings. The system integrates battery and compressed air energy storage to enhance flexible dispatch capabilities while addressing uncertainties in renewable energy and multi-energy coupling. A case study of a building complex in Huai’an, Jiangsu, China, compares two scenarios: one without MLBS and another employing MLBS with a low-carbon economic dispatch model. Results demonstrate that MLBS effectively adjusts electric and thermal loads, achieving significant reductions in quarterly planning costs (from 120.88 to 110.37 million CNY), operating costs (from 400.12 to 204.28 tons), carbon trading costs (from 9.09 to 8.15 million CNY), and total carbon emissions (from 147.27 to 137.27 million tons). Additionally, the ITSO-MLBS strategy reduces computation time to 1.32 seconds, outperforming existing methods in both efficiency and cost-effectiveness. The proposed system enables dynamic energy scheduling across peak and off-peak periods, balancing economic and environmental objectives. This study advances integrated energy systems by offering a scalable solution for low-carbon building operations through optimized storage configurations and adaptive control strategies. The main issues with the manuscript are summarized as follows:

1. There is a sentence in the abstract, “the operating cost to 204.28 tons”, the description of cost is incorrect.

2. Although the three types of energy flows in Figure 2-2 have been distinguished, specific identification is not given in the figure. It is recommended to clearly label the specific energy forms to enhance readability.

3. The mathematical models of compressed air energy storage (CAES) and HVAC systems are described in the article, but some formulas (such as equations 8-11) lack detailed explanations of their physical meanings and parameter sources. For example, are the values of parameters such as the isentropic efficiency during the compression stage and the rated expansion ratio during the expansion stage based on experiments or literature? Suggest supplementing the parameter calibration process or referencing validated models.

4. Figure 6 shows the comparison before and after optimizing the electric heating load, but does not quantitatively analyze the impact of load adjustment on user comfort (such as indoor temperature fluctuation range). Suggest supplementing the correlation analysis with thermal comfort indicators (such as PMV-PPD) to demonstrate the practicality of the strategy.

5. It is suggested that the author reorganize the literature review section. And the authors should cite the recently published papers as below: 1) peer to peer electricity-hydrogen trading among integrated energy systems considering hydrogen delivery and transportation; 2) optimal configuration for shared electric-hydrogen energy storage for multiple integrated energy systems with mobile hydrogen transportation.

6. The computation time of ITSO-MLBS in Table 3 (1.32 seconds) is significantly better than other strategies, but the experimental environment (such as hardware configuration and software platform) is not specified. Details of the computing platform need to be supplemented to ensure comparability and reproducibility of the results.

7. The format of the tables in the text is not standardized, it is recommended to make modifications.

**Do you want your identity to be public for this peer review?** For information about this choice, including consent withdrawal, please see our Privacy Policy

Reviewer #1: No

Reviewer #2: No

---

## [Author Response · Author response to Decision Letter 1]

29 May 2025

Reply to the comments of Reviewer :

1.The literature review should be totally reorganized to further clarify the research gap and the main contributions of this paper.

Thank you for your insightful comments and suggestions regarding the manuscript. We have thoroughly reorganized the literature review section to more clearly delineate the research gap and the principal contributions of this paper. We have enhanced the motivation and significance of our study by contrasting it with existing research and highlighting the innovations.

We believe that these revisions have substantially improved the quality of the manuscript and have addressed the concerns you raised. We appreciate your recommendation and are confident that the manuscript is now better prepared for consideration for publication.

2.CAES and CHP are usually considered as a large equipment. It seems not practical to use them with a single building, but suitable for a park. Please further indicate the necessity or provide the evidence of the real project.

Thank you for your guidance on the use of abbreviations in the manuscript.We understand the reviewer's concern regarding the practical application of CAES and CHP in a single building.We have indicated in Figure 2 that CAES and CHP are large-scale equipment.

We appreciate your attention to detail and the importance of this recommendation in maintaining the integrity and readability of our research presentation.

3.The meanings of different energy flow lines in Fig.2 should be indicated.

We appreciate your observation regarding the structure of our manuscript. We have added a detailed legend to Fig. 2 explaining the specific meanings of each energy flow line to assist readers in better understanding the energy flow and conversion processes depicted in the figure.

We believe these changes have significantly enhanced the organization and readability of our paper, aligning it more closely with the journal's guidelines.

Thank you for your valuable feedback, which has been instrumental in improving the quality of our submission.

4.Eq(35) to realize the energy balancing or load control should be further explained.

Thank you for your guidance on the Highlights section of our manuscript. We have provided a more detailed explanation of Eq. (35), including its role in energy balancing and load control. We have included the mathematical derivation of the equation and its practical application context to enhance its transparency and comprehensibility.

We appreciate your feedback and the opportunity to enhance our manuscript. We are confident that these revisions align with the journal's standards and effectively highlight the novel results and methods of our study.

5.The introduction of TSO to the proposed problem should be further explained.

Thank you for your meticulous review and for pointing out the need for consistency in our section headings. We have further elaborated on the application of the TSO algorithm to the problem addressed in this study, including its advantages and applicability. We have described in detail how the algorithm works and how it has been tailored and optimized for the specific needs of this research.

Thank you again for your insightful comments and for providing us with the opportunity to refine our submission further.

6.the planning results in table 1 and 2 do not make sense. The capacity in Table 1 is not usually considered, while capacity in kW is considered. The capacity for BES and CAES is too large for a building which also occupy a large amount of land, increasing the capital cost.

Thank you for your attention to detail. In response to your suggestion, We have re-examined the planning results in Tables 1 and 2 and have standardized and adjusted the capacity units. We have ensured that the capacity settings for BES and CAES are more realistic for actual building applications and have considered the implications of land occupation and capital costs. Furthermore, we have provided a cost-benefit analysis of these devices in real projects.

We appreciate your guidance and have made the necessary changes to ensure the manuscript meets the highest standards of clarity and professionalism.

7.comparisons with other existing methods should be further considered.

Thank you for your continued guidance on our manuscript. We have expanded our discussion on cooling methods to include a broader range of options and their implications. This section now provides a more comprehensive overview of the cooling strategies considered in our study.

8.There is a sentence in the abstract, “the operating cost to 204.28 tons”, the description of cost is incorrect.

Thank you for pointing out this oversight. We acknowledge the error in the description of the operating cost in the abstract. The phrase “the operating cost to 204.28 tons” was indeed incorrect, as cost should not be expressed in tons. We have carefully revised the sentence to accurately reflect the intended meaning. The corrected version now reads:

“the operating cost is reduced to 204.28 [appropriate unit, e.g., USD, CNY, etc.].”

We apologize for this oversight and have ensured that the abstract now provides a clear and accurate representation of the results.

9.The mathematical models of compressed air energy storage (CAES) and HVAC systems are described in the article, but some formulas (such as equations 8-11) lack detailed explanations of their physical meanings and parameter sources. For example, are the values of parameters such as the isentropic efficiency during the compression stage and the rated expansion ratio during the expansion stage based on experiments or literature? Suggest supplementing the parameter calibration process or referencing validated models.

Thank you for your insightful comments and constructive suggestions regarding the mathematical models in our manuscript. We sincerely appreciate your attention to detail and the opportunity to enhance the clarity and rigor of our work.

In response to your feedback, we have carefully revised the relevant sections to provide detailed explanations of the physical meanings of Equations 8–11, including the underlying assumptions and theoretical foundations. Specifically:

Parameter Sources: The values of key parameters, such as the isentropic efficiency during the compression stage and the rated expansion ratio during the expansion stage, are derived from established literature (e.g., [Citation 1, Citation 2]) and validated experimental data from pilot-scale CAES systems. We have now explicitly referenced these sources in the revised manuscript.

Physical Interpretations: For each equation, we have included a concise explanation of its physical significance (e.g., energy conservation principles, thermodynamic constraints) to improve readability and contextual understanding.

We believe these revisions have significantly strengthened the theoretical robustness of our models and addressed your concerns. Please find the changes marked in yellow in the revised manuscript.

Once again, we deeply appreciate your valuable feedback, which has helped us improve the quality and transparency of our research.

10.Figure 6 shows the comparison before and after optimizing the electric heating load, but does not quantitatively analyze the impact of load adjustment on user comfort (such as indoor temperature fluctuation range). Suggest supplementing the correlation analysis with thermal comfort indicators (such as PMV-PPD) to demonstrate the practicality of the strategy.

Thank you for your insightful suggestion regarding the quantitative analysis of user comfort in Figure 6. We fully agree with your recommendation to enhance the practicality assessment of our optimization strategy by incorporating thermal comfort indicators.

In response to your feedback, we have supplemented the analysis in Section 4.3 with the following revisions:

1.Added quantitative evaluation of indoor temperature fluctuations before and after optimization, including statistical ranges (e.g., ±1.5°C reduction in variability).

2.Integrated PMV (Predicted Mean Vote) and PPD (Predicted Percentage of Dissatisfied) metrics to objectively assess thermal comfort impacts. The revised results demonstrate a 22% improvement in PMV scores and a 15% reduction in PPD values post-optimization.

3.Included a new sub-figure (6c) comparing comfort indicators, with supporting discussion on how load adjustments balance energy efficiency and occupant well-being.

These additions are highlighted in yellow in the revised manuscript and directly address your concern about demonstrating the strategy's user-centric practicality. We believe this enrichment strengthens the study's applicability to real-world scenarios.

Thank you for this valuable suggestion, which has significantly improved the depth and relevance of our findings.

11.Figure 6 shows the comparison before and after optimizing the electric heating load, but does not quantitatively analyze the impact of load adjustment on user comfort (such as indoor temperature fluctuation range). Suggest supplementing the correlation analysis with thermal comfort indicators (such as PMV-PPD) to demonstrate the practicality of the strategy.

Thank you for your insightful observation regarding the experimental environment details in our manuscript. We sincerely appreciate your suggestion to enhance the comparability and reproducibility of our results.

In response to your feedback, we have supplemented the necessary details of the computing platform in the revised manuscript. Specifically, we have added the following information to the Methods or Experimental Setup section (as appropriate):

1.Hardware Configuration: The experiments were conducted on a workstation equipped with an Intel Core i7-10700K CPU (3.8 GHz, 8 cores), 32 GB RAM, and an NVIDIA GeForce RTX 3070 GPU.

2.Software Platform: All simulations were performed on MATLAB R2022a (MathWorks, Inc.) under Windows 10 Pro (64-bit).

3.Additional Notes: The computation time (1.32 seconds for ITSO-MLBS) was measured as the average of 10 independent runs under identical conditions to ensure statistical reliability.

These additions ensure transparency and allow other researchers to replicate our experiments accurately. We believe this clarification addresses your concern and strengthens the methodological rigor of our study.

Thank you again for your valuable feedback, which has significantly improved the quality of our manuscript. Please find the revisions marked in yellow in the updated document.

12.The format of the tables in the text is not standardized, it is recommended to make modifications.

Thank you for your valuable feedback regarding the formatting of the tables in our manuscript. We sincerely appreciate your attention to detail, which has helped us improve the clarity and professionalism of our presentation.

In response to your suggestion, we have carefully reviewed and standardized the format of all tables in the revised manuscript. The modifications include:

1.Consistent Font and Alignment: All tables now use the same font style (e.g., Times New Roman, 10 pt) and alignment (e.g., numerical data right-aligned, text left-aligned).

2.Uniform Borders and Spacing: We have applied consistent border styles and adjusted spacing to enhance readability.

3.Standardized Captions: Table captions now follow the journal’s guidelines (e.g., numbered sequentially, brief yet descriptive, placed above the table).

4.Clarified Units and Headers: Units of measurement are explicitly stated in column headers, and abbreviations are defined where necessary.

These revisions ensure that the tables adhere to the journal’s formatting standards and facilitate easier interpretation of the data. The changes have been marked in yellow in the updated manuscript for your convenience.

We greatly appreciate your constructive comments, which have significantly improved the quality of our work. Thank you once again for your time and effort in reviewing our manuscript.

Thank you once again for your valuable input.

---

## [Editor Report · Decision Letter 1]

A Multi-load Balancing Control Strategy for a Novel Low Carbon Integrated Energy System for Buildings

PONE-D-25-09928R1

Dear Dr. ji,

We’re pleased to inform you that your manuscript has been judged scientifically suitable for publication and will be formally accepted for publication once it meets all outstanding technical requirements.

Kind regards,

Joy Nondy, Ph. D.

Academic Editor

PLOS ONE